# Are Microalgae New Players in Nitrous Oxide Emissions from Eutrophic Aquatic Environments?

**Laura Teuma [1], Emanuel Sanz-Luque [2] , Benoit Guieysse [1] and Maxence Plouviez [1],***

[1] School of Food and Advanced Technology, Massey University, Palmerston North 4410, New Zealand; laura.teuma@gmail.com (L.T.); b.j.guieysse@massey.ac.nz (B.G.)

[2] Department of Biochemistry and Molecular Biology, University of Cordoba, 14071 Cordoba, Spain; q92salue@uco.es

\* Correspondence: m.plouviez@massey.ac.nz

**Abstract:** Anthropogenic activities cause the introduction of nitrogen (N) into aquatic environments where these N inputs drive the biological synthesis of nitrous oxide ($N_2O$), a potent and ozone-depleting greenhouse gas. To assess the significance of $N_2O$ emissions to climate change, the Intergovernmental Panel on Climate Change (IPCC) estimates indirect $N_2O$ emissions from rivers, lakes, and estuaries by multiplying the amounts of N received by these ecosystems with specific emission factors. Interestingly, the IPCC recently increased the $N_2O$ emission factor associated with wastewater discharge into "nutrient-impacted (eutrophic) aquatic receiving environments" nearly four times based on experimental evidence of high $N_2O$ emissions from N-receiving eutrophic ecosystems. As microalgae can produce $N_2O$, these organisms may contribute to the $N_2O$ emissions frequently reported in eutrophic aquatic bodies. If that is the case, estimating $N_2O$ emissions solely based on nitrogen inputs to water bodies might lead to inaccurate $N_2O$ budgeting as microalgae growth is often limited by phosphorus in these environments. Establishing the significance of microalgal $N_2O$ synthesis in eutrophic environments is, therefore, critical and may lead to considerable changes on how to budget and mitigate $N_2O$ emissions and eutrophication.

**Keywords:** nitrous oxide; eutrophication; microalgae; greenhouse gas budgeting; phosphorus

## 1. Introduction

The natural and/or anthropogenic (e.g., farm runoff) introduction of nitrogen (N) and phosphorus (P) into water bodies can cause the excessive growth of primary producers (i.e., plants) such as microalgae. This phenomenon, known as eutrophication, affects aquatic ecosystems globally [1–3] and is now considered a global environmental issue because of its increased occurrence [3]. The overgrowth of microalgae through massive blooms disturbs aquatic ecosystems in many ways. When actively growing, the blooms can harm the ecosystem by producing toxins, preventing light penetration, and modifying the water pH [4–6]. When dying, the blooms are decomposed by microbes severely depleting dissolved oxygen to levels that cannot support most life [7,8]. Unfortunately, another potential issue related to eutrophication has potentially been overlooked: the synthesis and emission of the greenhouse gas nitrous oxide ($N_2O$).

$N_2O$ is a globally significant ozone-depleting pollutant [9] and greenhouse gas (GHG) with a global-warming potential 273-times higher than $CO_2$ on a 100-year time scale [10]. Global $N_2O$ emissions from oceans, inland, and coastal waters were estimated at 4300 kt $N-N_2O \cdot year^{-1}$ from 2007 to 2016, and 14% of these emissions were caused by anthropogenic activities [11]. The Intergovernmental Panel on Climate Change (IPCC) assumes anthropogenic $N_2O$ emissions are indirectly caused by the introduction of anthropogenic N inputs into inland and coastal waters: (1) The leaching and runoff of N from agricultural soils; (2) The volatilization of N from land and its redeposition on water surfaces; (3) The discharge of N-rich wastewater into water bodies. A critical assumption behind the

current methodology is that $N_2O$ synthesis is mostly caused by bacterial nitrification and denitrification and that, consequently, $N_2O$ emissions are directly correlated to N inputs. Challenging this view, eutrophic aquatic environments are characterized by the abundance of microalgae whose growth is not necessarily linked (or linearly correlated) to the N inputs received and whose ability to produce $N_2O$ has now been clearly demonstrated [12–17]. Microalgae species from the Bacillariophyta, Chlorophyta, and Cyanobacteria have indeed been shown to synthesize $N_2O$ in the laboratory [13,18] and significant $N_2O$ emissions have been reported during microalgae cultivation outdoors [12,19,20].

Based on recent evidences, this article challenges the accuracy of current $N_2O$ budgeting methodologies in view of the potential significance of microalgal $N_2O$ synthesis and the impact of P pollution on this potentially new source. Importantly, the aim of this article is to raise awareness about the potential global impact of the ability of microalgae to synthesize $N_2O$. Acknowledging and investigating microalgal $N_2O$ emissions from eutrophic environments is of paramount importance to establish the role microalgae play in $N_2O$ emissions from these aquatic ecosystems and if monitoring and greenhouse gases budgeting methodologies needs to be revised. Noteworthy, the terms hyperoxia, normoxia, hypoxia, and anoxia will be used to describe oversaturated, normal, low, and absent levels of dissolved oxygen in aquatic bodies, respectively. In addition, while microalgae are taxonomically eukaryotic phototrophs, cyanobacteria are often considered as microalgae in the literature and this broader definition will be used in this article for simplicity.

## 2. $N_2O$ Inventory for Aquatic Environments

### 2.1. Current Methodology

Figure 1 depicts the Tier 1 methodology recommended by the IPCC to estimate anthropogenic $N_2O$ emissions from aquatic environments. These indirect $N_2O$ emissions are estimated by multiplying the N loads predicted to be received with emission factors specific to emitting activities and/or receiving environments using a three-tiered approach: (1) The Tier 1 method calculates GHG emissions using default emission factors and, when applicable, default partitioning factors to estimate N loads; (2) The Tier 2 method uses country-specific emission factors and partitioning factors; (3) The Tier 3 method uses country-specific models and data [21]. Tier 1 emission factors of 0.01 and 0.011 kg $N$–$N_2O\cdot$kg N input$^{-1}$ are currently recommended to estimate indirect $N_2O$ emissions from aquatic environments receiving N via atmospheric deposition ($EF_4$) and from agricultural runoff and leaching ($EF_5$), respectively [21]. A Tier 1 emission factor of 0.005 kg $N$–$N_2O\cdot$kg $N^{-1}$ is also recommended to estimate indirect $N_2O$ emissions from aquatic environments receiving N from domestic and industrial wastewater effluents [22]. These Tier 1 emission factors were calculated based on experimental measurements of $N_2O$ emissions and N inputs in various aquatic ecosystems. Indirect aquatic $N_2O$ emissions due to N inputs from agricultural soils were, thus, calculated based on data from 106 studies [23]. The emission factor used to compute $N_2O$ emissions from soils, lakes, and other waters ($EF_4$) has the same value as the emission factor used to estimate $N_2O$ emissions from the direct application of fertilizers because the deposition of N on land and water surfaces is considered to be equivalent to the application of fertilizers [21]. $N_2O$ emissions resulting from the discharge of N-laden wastewater effluents into water bodies are estimated using a specific emission factor for effluent discharge ($EF_{EFFLUENT}$). This factor was calculated based on the average of the ratios of dissolved $N$–$N_2O$- concentration to N concentration found in the literature. The default Tier 1 $EF_{EFFLUENT}$ value was calculated using data from 62 well-oxygenated environments [22].

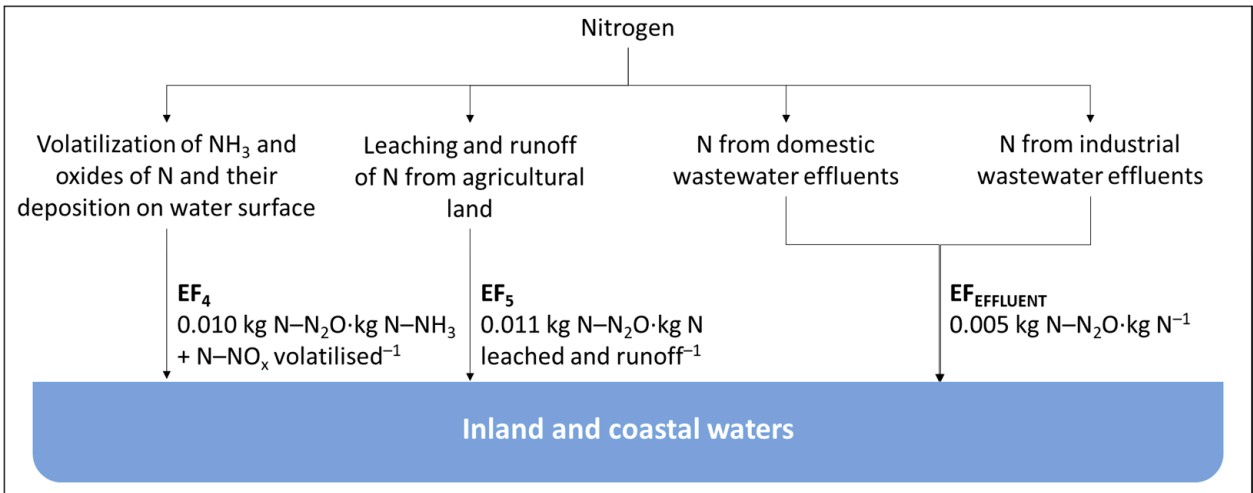

**Figure 1.** Current emission factors used for the different N fluxes causing indirect anthropogenic N$_2$O emissions from aquatic environments.

### 2.2. Limitations to the Current Approach

Numerous species of bacteria, archaea, fungi, and microalgae can produce N$_2$O via processes known as denitrification, nitrification, and nitrogen uptake from nitrates and nitrites [24–27].

**Bacterial N$_2$O synthesis** can occur during nitrification, denitrification, coupled nitrification–denitrification, nitrifier–denitrification, and anaerobic ammonia oxidation in the benthic zone of different aquatic ecosystems [6,28–31]. During nitrification, ammonia is oxidized into hydroxylamine (NH$_2$OH) by the enzyme ammonia monooxygenase and NH$_2$OH is further oxidized to nitrite (NO$_2^-$) by hydroxylamine oxidoreductase. During this process, N$_2$O can be formed from the spontaneous chemical decomposition of NH$_2$OH or NO$_2^-$ [28,30,32]. Denitrifying bacteria are responsible for N$_2$O production during partial denitrification under hypoxia or anoxia. Denitrification is a respiratory pathway during which nitrate (NO$_3^-$) is first reduced to NO$_2^-$ by nitrate reductase (NR) and NO$_2^-$ is then reduced to nitric oxide (NO) by nitrite reductase (NiR). NO is, in turn, reduced to N$_2$O by nitric oxide reductase (NOR) and N$_2$O is finally reduced to nitrogen (N$_2$) by nitrous oxide reductase [33]. During denitrification, N$_2$O production increases with the presence of O$_2$ and at low pH as these conditions inhibit nitrous oxide reductase (N$_2$OR) activity.

**Fungal N$_2$O synthesis** occurs via denitrification, as described above for bacteria, and co-denitrification catalyzed by the fungal NOR when a nitrogen co-substrate (e.g., NH$_4^+$, amino acids, or urea) is available [30,34,35].

**Archaeal N$_2$O synthesis** occurs during the first step of nitrification and this pathway significantly contributes to N$_2$O emissions from oceans [36–38].

**Microalgal N$_2$O synthesis** occurs via the successive reduction of NO$_3^-$ into NO$_2^-$ and NO then converted into N$_2$O via various putative pathways as further detailed in Section 3.2.

As illustrated in Figure 1, the IPCC currently considers that N$_2$O emissions increase linearly with N inputs because it assumes N$_2$O production mainly depends on the N$_2$O yields of bacterial nitrification and denitrification [21,22]. Webb et al. [39], however, argued that indirect N$_2$O emissions from agricultural surface waters were overestimated using this approach due to the impacts of factors such as the hydraulic retention time (HRT) and substrate availability (i.e., N$_2$O emissions do not linearly increase with N input). N$_2$O production and consumption have indeed been shown to be influenced by various parameters in inland and coastal waters. The size and morphology of water bodies, especially water depth, and the HRT of the water impact the biological productivity of rivers [40], lakes [41], estuaries [42], and oceans [43] through interactions between the water, nutrient cycling, and/or microorganisms [41–43]). A long HRT means that

phytoplankton suspended in water have more time to uptake nutrients before being flushed out [44]. Miao et al. [45] also noticed a significant seasonality and spatial variation of $N_2O$ emissions from lake Chaohu, China, and suggested that these emissions were under the control of factors such as the transfer velocity of $N_2O$ from water to the atmosphere, water temperature, and dissolved oxygen concentration. For example, high temperatures reduce the solubility of dissolved oxygen and can, therefore, promote denitrification [46].

Another methodological limitation to estimate $N_2O$ emissions is that the same emission factor is used for different environments. For example, the value of 0.0026 kg $N–N_2O \cdot$kg N leached and runoff$^{-1}$ is used for both reservoirs and rivers despite significantly different emissions being documented for reservoirs (0.17–0.44 kg $N–N_2O \cdot$kg N leached and runoff$^{-1}$) and rivers (0.004–0.005 kg $N–N_2O \cdot$kg N leached and runoff$^{-1}$, [47]). The observed differences could be explained by the impact of HRT on biological $N_2O$ synthesis [47]. Mulholland et al. [48] used nitrogen stable isotope tracing to study the nitrogen removal in 72 streams located in the United States and Puerto Rico and evidenced a reduction in the emission rate using the N load. As a consequence from these experimental observations, nonlinear models for $N_2O$ emissions from streams and rivers considering environmental parameters such as size, morphology, and climate have been proposed in the literature [40,49].

The IPCC is constantly improving its methodology based on new data and findings and it is, therefore, critical to challenge its assumptions and provide potential solutions for improvement. The discussion below specifically challenges the assumption that bacterial mechanisms are the only significant sources of $N_2O$ in eutrophic environments.

## 3. Potential Significance of Microalgal $N_2O$ Synthesis

### 3.1. A case for Microalgal $N_2O$ Emissions

High $N_2O$ emissions have been reported during algal blooms in various ecosystems (Tables 1 and 2). Interestingly, when focusing on lakes, various authors have attributed these emissions to the decay of microalgae causing bacterial denitrification under hypoxia or anoxia [50–52]. However, high $N_2O$ emissions recorded under normoxia have been correlated to chlorophyll *a* concentration, a proxy for microalgae biomass concentration [2,53,54], which suggests bacterial denitrification may not be the mechanism involved in these oxic environments. In addition, results from Teuma [55] showed that the addition of $NO_2^-$ to microalgae-rich normoxic lake samples incubated with or without antibiotics produced $N_2O$ at similar rates during a 24 h period. While this needs to be confirmed in the field, these results also suggest that bacteria are not necessarily the main producers of $N_2O$ in lakes and potentially other ecosystems. The ability of microalgae to synthesize $N_2O$ combined with their ubiquity in eutrophic ecosystems, therefore, mandate investigating if these microorganisms indeed play a significant role in $N_2O$ emissions. Unfortunately, current methodologies for GHG monitoring are not designed to cope with the fickle nature of microalgal $N_2O$ emissions because the rate of microalgal $N_2O$ synthesis indeed rapidly fluctuates (minutes) with solar irradiance depending on cloud cover [13,19,56]. $N_2O$ monitoring from aquatic ecosystems should, therefore, prevent artificial shading of the microalgae and involve frequent sampling. Water depth and flow (i.e., turbulence) should be considered during monitoring as turbulence resuspend particles (e.g., silt, clay, and microorganisms) reducing light penetration in water. Thus, the growth of benthic primary producers is limited to low-depth areas with clear water and low turbulence [44]. In many lakes, estuaries and coastal waters, photosynthetic activity mainly occurs near the surface as the primary producers themselves shade the water column below. The occurrence of algal blooms can also increase in poorly mixed water and during sustained water stratification. When turbulence becomes excessive, phytoplankton growth is reduced or can even stop due to cell damage caused by shear [57] while thermal stratification increases the risk of algal bloom because phytoplankton can be trapped in a nutrient-rich layer near the surface, where light is also available [44].

### 3.2. N$_2$O Emissions from Microalgae Ecosystems

The ability of microalgae to synthesize and emit N$_2$O is now well established [12–17] (Table 1) and N$_2$O emissions from various aquatic environments harboring microalgae have been repeatedly reported in the literature (Table 2). Based on the limited amount of data available, several authors have predicted that global N$_2$O emissions from eutrophic lakes, that only started to be considered in the 2019 IPCC methodology refinement, could represent 52 to 211 Tg CO$_2$-eq·year$^{-1}$ [2,12,19], which represent 3 to 12% of all direct anthropogenic N$_2$O emissions from agriculture, the largest contributor to anthropogenic N$_2$O emissions [11]. Based on the data presented in Table 2, similar N$_2$O fluxes were reported from lakes, coastal waters, and oceans. There is, therefore, a need to better understand microalgal N$_2$O synthesis and assess its potential significance, particularly when algae blooms are occurring more frequently and globally due to anthropogenic activities [2,42,58–61].

**Table 1.** Algae species and N$_2$O fluxes reported during microalgae cultivation (modified from Plouviez et al. [12]).

| N$_2$O Emissions from Laboratory culturess and Engineered Systems | | | | |
|---|---|---|---|---|
| **Alga Division** | **Algal Species** | **Ecosystem** | **N$_2$O Flux** | **Reference** |
| **Green microalgae** | *Chlorella vulgaris* | Laboratory assays | 109–1480 nmole·h$^{-1}$·g DW$^{-1}$ | [16] |
| | | Photobioreactor | 563–4134 nmole·h$^{-1}$·g DW$^{-1}$ | [16] |
| | | Photobioreactor | 9.60–38,000 nmole·m$^{-2}$·h$^{-1}$ | [56] |
| | | Raceway pond | 2–5685 nmole·h$^{-1}$·g DW$^{-1}$ | [62] |
| | *C. rubescens* | Laboratory assays | 1200–2500 nmole·h$^{-1}$·g DW$^{-1}$ | [14] |
| | *C. variabilis* | Laboratory assays | 300 µmole·L$^{-1}$·h$^{-1}$ | [17] |
| | *Coelastrum* sp. | Laboratory assays | 560–1100 nmole·h$^{-1}$·g DW$^{-1}$ | [14] |
| | *Chlorococcum vacuolarum* | Laboratory assays | 150–290 nmole·h$^{-1}$·g DW$^{-1}$ | [14] |
| | *Neochloris* sp. | Photobioreactor | 50–14,200 nmole·m$^{-2}$·h$^{-1}$ | [56] |
| | *Scenedesmus dimorphus* | Laboratory assays | 6–73 nmole·h$^{-1}$·g DW$^{-1}$ | [63] |
| | *S. obliquus* | Laboratory assays | 0–1000 nmole·h$^{-1}$·g DW$^{-1}$ | [14] |
| | *Chlamydomonas reinhardtii* | Laboratory assays | 7.5–74 nmole·h$^{-1}$·g DW$^{-1}$ | [13] |
| | | Laboratory assays | 54 µmole·L$^{-1}$·h$^{-1}$ | [17] |
| | *Coccomyxa subellipsoidea* | Laboratory assays | 225 µmole·L$^{-1}$·h$^{-1}$ | [17] |
| | *Tetraselmis subcordiformis* | Laboratory assays | 188 µmole·L$^{-1}$·h$^{-1}$ | [17] |
| **Eustigmatophyceae** | *Nannochloropsis oculata* | Laboratory assays | 0.98 nmole·L$^{-1}$·h$^{-1}$ | [64] |
| **Diatoms** | *Skeletonema marinoi* | Laboratory assays | 0.039–0.31 nmole·h$^{-1}$·aggregate$^{-1}$ | [65] |
| | *Thalassiosira weissflogii* | Laboratory assays | 0.087–0.3 nmole·L$^{-1}$·h$^{-1}$ | [66] |
| | *Staurosira* sp. | Raceway pond | −212.5–316.7 nmole·m$^{-2}$·h$^{-1}$ | [67] |
| **Cyanobacteria** | *Aphanocapsa 6308* | Laboratory assays | 0–1500 nmole·h$^{-1}$·g DW$^{-1}$ | [15] |
| | *Aphanocapsa 6714* | Laboratory assays | 0–5700 nmole·h$^{-1}$·g DW$^{-1}$ | [15] |
| | *Nostoc* sp. | Laboratory assays | 0–1500 nmole·h$^{-1}$·g DW$^{-1}$ | [15] |
| | *Microcystis aeruginosa* | Laboratory assays | 0–198.9 nmole·h$^{-1}$·g DW$^{-1}$ | [18] |

**Table 2.** $N_2O$ emissions from natural environments reported from studies acknowledging algal involvement in the $N_2O$ emissions (modified from Plouviez et al. [12]).

| N₂O Emissions from Aquatic Ecosystems | | | |
|---|---|---|---|
| **Ecosystem** | **N$_2$O Flux** | **O$_2$ Conditions** [1] | **Reference** |
| Ocean | 115 nmole·m$^{-2}$·h$^{-1}$ | Normoxic | [68] |
| Ocean | 409 nmole·m$^{-2}$·h$^{-1}$ | Hypoxic | [69] |
| Coastal wetland | 125–228 nmole·m$^{-2}$·h$^{-1}$ | Anoxic and hypoxic | [70] |
| Ocean | 123–132% saturation | Normoxic | [71] |
| Lake (including eutrophic ones) | 300–700 nmole·m$^{-2}$·h$^{-1}$ | From anoxic to normoxic | [72] |
| Ocean | 88 nmole·m$^{-2}$·h$^{-1}$ | Not specified | [73] |
| Lake (eutrophic) | 357–2450 nmole·m$^{-2}$·h$^{-1}$ | Not specified | [74] |
| Lake | 0–10,057 nmole·m$^{-2}$·h$^{-1}$ | Oxic | [75] |
| Lake (eutrophic) | 46–230 nmole·m$^{-2}$·h$^{-1}$ | From anoxic to hypoxic | [54] |
| Lake | 12.5–2233 nmole·m$^{-2}$·h$^{-1}$ | Normoxic and Hyperoxic | [45] |

[1] Anoxic conditions occur when dissolved oxygen concentration in freshwater ([$O_2$]) is inferior to 3 μM, hypoxic conditions when 3 μM < [$O_2$] < 200 μM, normoxic conditions when 200 μM < [$O_2$] < 400 μM, and hyperoxic conditions when [$O_2$] > 400 μM [76].

Microalgal $N_2O$ synthesis requires NO, which is mainly synthesized from $NO_2^-$ by the enzyme NR, and the subsequent reduction of NO into $N_2O$ [77–79]. In the chloroplast, NO reduction into $N_2O$ is mediated by flavodiiron proteins (FLVs) using electrons from photosynthesis [17,80]. In the mitochondria, the NO reductase CYP55 has been demonstrated to carry out NO reduction to $N_2O$ [13]. In addition to FLVs and CYP55, *Chlamydomonas* possess four Hybrid Cluster Proteins (HCPs) found in an extensive range of prokaryote and eukaryote organisms [81]. While the physiological functions of HCPs remain uncertain in eukaryotic microalgae, HCPs may be responsible for $N_2O$ production under anaerobic conditions in bacteria [82]. The *Chlamydomonas* genome also contains a gene homologous to the bacterial and fungal genes encoding a copper-containing nitrite reductase [27], which is absent in most eukaryotes [83]. The role of this enzyme has not been described yet, but its presence suggests another potential pathway for NO synthesis from $NO_2^-$ in the mitochondria. Bellido-Pedraza et al. [27] estimated that nearly one third of the 100 photosynthetic microorganisms described in genomic databases contain at least one of the proteins involved in $N_2O$ synthesis in *Chlamydomonas* (NirK, CYP55, FLVs, and HCP), including the widely distributed and dominant cyanobacterium in freshwater ecosystems *Microcystis aeruginosa* [18]. There is, therefore, clear evidence for the existence of several broadly distributed $N_2O$ synthesis pathways in microalgae. This, in turn, suggests that microalgal $N_2O$ emissions should occur in many algae-rich ecosystems and under many conditions.

## 4. Nitrogen, the Perfect Culprit for N₂O Emissions from Eutrophic Environments?

### 4.1. N₂O Emissions under Oxia

Based on "research published between 1978 and 2017 [. . .] indicating that higher $N_2O$ emissions occur when wastewater is discharged to nutrient-impacted (eutrophic) or hypoxic aquatic receiving environments", the IPCC recently increased the EF associated with wastewater discharge into "nutrient-impacted waters" from 0.005 to 0.019 kg N–$N_2O$·kg $N^{-1}$ [22]. The IPCC, however, postulates that the higher $N_2O$ emissions experimentally recorded in eutrophic waters are caused by bacterial $N_2O$ synthesis enhanced under hypoxic/anoxic conditions caused by light attenuation (due to microalgae proliferation) and microalgae decay. Consequently, the IPCC does not recommend to increase the emission factor used to compute $N_2O$ emissions associated with N runoffs from agriculture, stating that a "combination of reducing conditions and high organic loading [. . .] are unlikely to

exist in agricultural runoff systems" [22]. Challenging this approach, microalgae have been reported to produce $N_2O$ under normoxia and even hyperoxia, meaning that dissolved oxygen concentration might not be the most relevant parameter to determine if eutrophic environments generate high $N_2O$ emissions. As suggested above, further monitoring of $N_2O$ emissions from eutrophic ecosystems would, therefore, be critical to inform if the current IPPC assumptions yield accurate $N_2O$ emission estimates or if, instead, refinements in accounting methodologies are needed.

### 4.2. Possible Impact of Phosphorus Inputs on $N_2O$ Emissions in Eutrophic Ecosystems

While it is currently unknown how P availability influences $N_2O$ emissions in eutrophic aquatic environments, P availability has been reported to influence $N_2O$ emissions from soils [84–87], with higher emissions reported when P is added to P-depleted soils [84,85].

P inputs into aquatic ecosystems can trigger microalgae proliferation, sometimes in combination with N inputs [88–92]. Moreover, P accumulation in the sediments can cause P reintroduction into water, which can trigger an algal bloom in the absence of recent P input [93,94]. Nutrient stoichiometry must also be considered because the N:P ratio can influence microalgal population dynamics [95–97] and impact microalgal diversity [98]. These impacts of P, together with the ability of microalgae to produce $N_2O$, may mean that $N_2O$ emissions from eutrophic environments may not only be correlated to N inputs. Understanding the impact of both N and P inputs on algal $N_2O$ emissions may, therefore, be critical to accurately estimate $N_2O$ emissions from eutrophic ecosystems. The potential impact of P could even trigger a paradigm shift on how $N_2O$ emissions are mitigated as this may require the limitations of both N and P inputs to water bodies [92].

### 4.3. Possible Impact of Micronutrients Inputs and Microbial Interactions in Eutrophic Ecosystems

While N and P are generally driving the level and duration of blooms in aquatic ecosystems, micronutrients such as metallic ions (e.g., Fe) may also be critical. Micronutrients are essential for microalgal growth [99] and these nutrients can influence microalgal diversity and/or trigger blooms in natural aquatic environments [100–102]. In addition, micronutrients such as copper have been shown to influence the activity of denitrifiers' nitrous oxide reductase enzymes [103]. As these potential impacts have not been characterized in microalgae, determining how and/if micronutrients influence microalgal $N_2O$ emissions, therefore, deserves careful investigation.

Another field of research requiring consideration is the impact of microalgal–microbial interactions on $N_2O$ emissions. In natural environments, microbes form complex relationships (symbiotic or nonsymbiotic) with other microorganisms involving nutrient or growth factor exchanges, quorum sensing mediation, and/or episodic parasitism/killing [104,105].

### 4.4. Are Microalgae New Players in Nitrous Oxide Emissions from Eutrophic Aquatic Environments?

To this date, it is still unclear whether microalgae contribute significantly to the $N_2O$ emissions reported from eutrophic aquatic ecosystems, especially for estuaries and coastal ecosystems. However, microalgae have been shown to synthesize $N_2O$ in various settings and preliminary assays suggested their involvement in $N_2O$ emissions in lake samples [55]. Further field data and/or microcosms are, therefore, needed to confirm these findings in many other aquatic ecosystems. Considering the sensitivity of microalgal $N_2O$ synthesis, and the seasonal variability in $N_2O$ emissions reported in the field [43,74,106], long-term with wide spatial coverage and high sampling frequency monitoring of various microalgae-rich environments are, therefore, needed to improve the accuracy of $N_2O$ emissions from these systems. The use of genomics would also be critical to unravel the occurrence and ecological implications of microalgal $N_2O$ synthesis and the potential interplay between microbial $N_2O$ biosynthetic pathways in those environments.

## 5. Conclusions

The IPCC currently estimates $N_2O$ emissions from aquatic environments by assuming that bacterial nitrification and denitrification processes leading to $N_2O$ synthesis are linearly related to the N inputs received by the aquatic body assessed. Thus, the $N_2O$ emissions are calculated as a fraction of the N flux reaching the aquatic body defined as emission factors (EFs). This 'bacteria-centric' assumption that $N_2O$ emissions only depends on the N input received is, however, challenged by the ability of microalgae to bloom and produce $N_2O$ in response to P inputs or combined N and P inputs. Eutrophic aquatic environments are already known to be a higher source of $N_2O$ than oligo/mesotrophic environments and the IPCC recently acknowledged this fact by increasing the Tier 3 factor used to compute indirect $N_2O$ emissions from wastewater discharge into eutrophic and nutrient-impacted aquatic environments from 0.005 to 0.019 kg $N–N_2O$ emitted per kg of N received. These higher emissions from eutrophic environments are only considered at a Tier 3 level for wastewater discharge (e.g., not for indirect $N_2O$ emissions from eutrophic aquatic ecosystems receiving N inputs from agricultural N leaching and runoff). In addition, microalgae are still not considered as one of the potential causes of these $N_2O$ emissions and, therefore, emissions are still only computed from N inputs. Further monitoring to track the exact source(s) of the $N_2O$ emissions in eutrophic aquatic environments, i.e., lakes, rivers, estuaries, and coastal waters, is critical for the following reasons: (1) Past monitoring based on 'bacteria-centric' methodologies may have missed the contribution of other organisms such as microalgae; (2) Microbial and particularly 'microalgal-$N_2O$ activity' could be influenced by P (phosphorus supply) triggering eutrophication, meaning that $N_2O$ emissions from affected ecosystems could no longer be based solely on N-loadings (as currently done). A better understanding of the microbial pathways (and interplays) involved during $N_2O$ emissions in eutrophic environments could improve how $N_2O$ emissions are predicted and mitigated. It could also improve our knowledge and assessment of natural $N_2O$ emissions in aquatic environments.

**Author Contributions:** Writing—original draft preparation, L.T.; writing—review M.P., B.G. and E.S.-L.; editing, L.T. All authors have read and agreed to the published version of the manuscript.

**Funding:** This study was financially supported by internal funds from Massey University, New Zealand (i.e., Laura Teuma scholarship).

**Institutional Review Board Statement:** Not applicable.

**Informed Consent Statement:** Not applicable.

**Data Availability Statement:** Data presented in this article can be found in the cited literature.

**Conflicts of Interest:** The authors declare no conflict of interest.

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
