# Peer review of "Are Microalgae New Players in Nitrous Oxide Emissions from Eutrophic Aquatic Environments?"

_phycology, doi:10.3390/phycology3030023_

Round 1

Reviewer 1 Report

Review for the paper "Are microalgae new players in nitrous oxide emissions from eutrophic aquatic environments?" by Laura Teuma, Emanuel Sanz-Luque, Benoit Guieysse, Maxence Plouviez submitted to "Phycology".

General comment.

The manuscript discusses the principal role of phytoplankton as facilitators for nitrous oxide emissions - a subject of considerable interest and debate. It critically reviews the extant methodologies implicated in measuring potential emissions of N2O from aquatic microalgae, arguing that they routinely furnish biased N2O measurements due to inherent limitations. The authors offer data highlighting the potential significance of N2O synthesis by varied species of phytoplankton. The paper is generally well-constructed, free from perceptible scientific discrepancies. However, I believe there remain issues requiring attention before a final manuscript acceptance.

Major concerns:

The authors have narrowed N2O emission sources to eutrophic lakes, while certain estuarine ecosystems and shelf regions disproportionately influenced by river discharge may also harbour substantial phytoplankton populations. The total bulk of these populations, I contend, could constitute an important source of atmospheric N2O release. Thus, I recommend weaving in a paragraph discussing the role of estuarine and coastal marine ecosystems in the aggregate budget of N2O resultant from microalgal activity.

The focus in Section 4.2 should also embrace other essential biogenic and trace elements and compounds. For example, various microbes are known to release certain metabolites that facilitate the growth and expansion of phytoplankton populations. Elements such as Fe, Mn, and B have been identified as vital catalysts for microalgal proliferations across diverse aquatic environments.

I also suggest the authors to incorporate a consideration of seasonality in the emissions of N2O by phytoplankton inhabiting eutrophic, estuarine, shelf, and coastal ecosystems.

Specific remarks:

For line 142, "a" must be italicized in the term "chlorophyll a".

Minor

Author Response

Reviewer 1:

The manuscript discusses the principal role of phytoplankton as facilitators for nitrous oxide emissions - a subject of considerable interest and debate. It critically reviews the extant methodologies implicated in measuring potential emissions of N2O from aquatic microalgae, arguing that they routinely furnish biased N2O measurements due to inherent limitations. The authors offer data highlighting the potential significance of N2O synthesis by varied species of phytoplankton. The paper is generally well-constructed, free from perceptible scientific discrepancies. However, I believe there remain issues requiring attention before a final manuscript acceptance.

We thank Reviewer 1 for her/his appraisal and all the issues were addressed as described below:

  1. The authors have narrowed N2O emission sources to eutrophic lakes, while certain estuarine ecosystems and shelf regions disproportionately influenced by river discharge may also harbour substantial phytoplankton populations. The total bulk of these populations, I contend, could constitute an important source of atmospheric N2O release. Thus, I recommend weaving in a paragraph discussing the role of estuarine and coastal marine ecosystems in the aggregate budget of N2O resultant from microalgal activity.

We thank Reviewer 1 for this suggestion, and fully agree a computation of the contribution of microalgal N2O synthesis in estuaries and costal marine ecosystems would be very valuable. However, we feel there is too little data currently available to undertake such assessment. Besides, in contrast to lakes (Delstontro et al., 2018), no correlation between Chlorophyll a has been reported for those ecosystems. Consequently, we did not add an estimate, but we better highlighted the need for further research in those ecosystems.

Section 3.2 (L203-204): “Based on the data presented in Table 2, similar N2O fluxes were reported from lakes, coastal waters and oceans”.

Section 4.4 (L292-294): “To this date, it is still unclear whether microalgae contribute significantly to the N2O emissions reported from eutrophic aquatic ecosystems, especially for estuaries and coastal ecosystems.”

Section 5 (L320-322): “Further monitoring to track the exact source(s) of the N2O emissions in eutrophic aquatic environments i.e. lakes, rivers, estuaries and coastal waters is critical because…”

  1. The focus in Section 4.2 should also embrace other essential biogenic and trace elements and compounds. For example, various microbes are known to release certain metabolites that facilitate the growth and expansion of phytoplankton populations. Elements such as Fe, Mn, and B have been identified as vital catalysts for microalgal proliferations across diverse aquatic environments.

The following section was added L275-288: “4.3. Possible impact of micronutrients inputs and microbial interactions in eutrophic ecosystems”: “While N and P are generally driving the level and duration of blooms in aquatic eco-systems, micronutrients such as metallic ions (e.g. Fe) may also be critical. Micronutrients are essential for microalgal growth [99]and these nutrients can influence microalgal diversity and/or trigger blooms in natural aquatic environments [100–102]. In addition, micronutrients such as copper have been shown to influence the activity of denitrifiers`’ nitrous oxide reductase enzymes [103]. As these potential impacts have not been characterized in microalgae, determining how and/if micronutrients influence microalgal N2O emissions therefore deserves careful investigation.

Another field of research requiring consideration is the impact of microalgal-microbial interactions on N2O emissions. In natural environment microbes form complex relationships (symbiotic or non-symbiotic) with other microorganisms involving nutrient or growth factor exchanges, quorum sensing mediation, and/or episodic parasitism/killing [104,105].”

  1. I also suggest the authors to incorporate a consideration of seasonality in the emissions of N2O by phytoplankton inhabiting eutrophic, estuarine, shelf, and coastal ecosystems.

This was addressed L297-298 by inserting “and the seasonal variability in N2O emissions reported in the field”.

  1. For line 142, "a" must be italicized in the term "chlorophyll a".

The manuscript was corrected accordingly (L170)

Reviewer 2 Report

ABSTRACT

this section is too long; please comply with MDPI guidelines

INTRODUCTION

this section is too short; please increase it

At the end of this section, please add the goal of the paper, make it clear to avoid confusion for the readers

SECTION 2

fuse section 2.3 "A case for microalgal N2O emissions" with section 3 as a "short introduction" to section 3

SECTION 3

When possible, please simplify table 1, specially scientific names (Chlorella vulgaris, C. rubescens, C. variabilis, etc) and ecosystems

please break table 2 in two separate tables (lab conditions and aquatic ecosystems)

SECTION 4

Please, at the end of this section, give a clear and direct answer to the question that was launched at the beginning of the paper "Are microalgae new players in nitrous oxide emissions from 2 eutrophic aquatic environments?"

Author Response

We thank Reviewer 2 for her/his review and all the issues were addressed as described below:

  1. this section is too long; please comply with MDPI guidelines

The Abstract was shortened to 185 words as “Anthropogenic activities cause the introduction of nitrogen (N) into aquatic environments where these N inputs drive the biological synthesis of nitrous oxide (N2O), a potent and ozone depleting greenhouse gas. To assess the significance of N2O emissions to climate change, the Intergovernmental Panel on Climate Change (IPCC) estimates indirect N2O emissions from rivers, lakes, and estuaries by multiplying the amounts of N received by these ecosystems with specific emission factors. Interestingly, the IPCC recently increased the N2O emission factor associated with wastewater discharge into “nutrient-impacted (eutrophic) aquatic receiving environments” nearly four times based on experimental evidence of high N2O emissions from N-receiving eutrophic ecosystems. As microalgae can produce N2O these organisms may contribute to the N2O emissions frequently reported in eutrophic aquatic bodies. If that is the case, estimating N2O emissions solely based on nitrogen inputs to water bodies might lead to inaccurate N2O budgeting as microalgae growth is often limited by phosphorus in these environments. Establishing the significance of microalgal N2O synthesis in eutrophic environments is therefore critical and may lead to considerable changes on how to budget and mitigate N2O emissions and eutrophication.”

  1. this section is too short; please increase it.

The following subsections were added/expended:

L32-42 “The natural and/or anthropogenic (e.g. farm runoffs) introduction of nitrogen (N) and phosphorus (P) in water bodies can cause the excessive growth of primary producers (i.e. plants) such as microalgae. This phenomenon, known as eutrophication, affects aquatic ecosystems globally [1–3] and is now considered a global environmental issue because of its increased occurrence [3]. The overgrowth of microalgae through massive blooms, dis-turb aquatic ecosystems in many ways. When actively growing, the blooms can harm the ecosystem by producing toxins, preventing light penetration and modifying the water pH [4–6]. When dying, the blooms are decomposed by microbes severely depleting dissolved oxygen to levels that cannot support most life [7,8]. Unfortunately, another potential issue related to eutrophication has potentially been overlooked: the synthesis and emission of the greenhouse gas nitrous oxide (N2O).”

L57-59: “Microalgae species from the Bacillariophyta, Chlorophyta, Cyanobacteria have indeed been shown to synthesise N2O in the laboratory [13,18] and significant N2O emissions have been reported during microalgae cultivation outdoors [12,19,20].”

L63-73: “Importantly, the aim of this article is to raise awareness about the potential global impact of the ability of microalgae to synthesise N2O. Acknowledging and investigating microalgal N2O emissions from eutrophic environment is of paramount importance to establish the role microalgae play in N2O emissions from eutrophic aquatic ecosystems and if monitoring and greenhouse gases budgeting methodologies needs to be revised. Noteworthy, the terms hyperoxia, normoxia, hypoxia and anoxia will be used to describe oversaturated, normal, low and absent levels of dissolved oxygen in aquatic bodies respectively. In addition, while microalgae are taxonomically eukaryotic phototrophs, cyanobacteria are often considered as microalgae in the literature and this broader definition will be used in this article for simplicity.”

  1. At the end of this section, please add the goal of the paper, make it clear to avoid confusion for the readers

The following sentence was added L63-68: “Importantly, the aim of this article is to raise awareness about the potential global impact of the ability of microalgae to synthesise N2O. Acknowledging and investigating microalgal N2O emissions from eutrophic environment is of paramount importance to establish the role microalgae play in N2O emissions from eutrophic aquatic ecosystems and if monitoring and greenhouse gases budgeting methodologies needs to be revised.”

  1. SECTION 2. fuse section 2.3 "A case for microalgal N2O emissions" with section 3 as a "short introduction" to section 3

Section 2.3 has been moved under section 3 as requested.

  1. SECTION 3. When possible, please simplify table 1, specially scientific names (Chlorella vulgaris, C. rubescens, C. variabilis, etc) and ecosystems please break table 2 in two separate tables (lab conditions and aquatic ecosystems)

The manuscript was simplified accordingly: Scientific named were simplified as much as possible and the Table was reformatted into 2 new Tables.

  1. SECTION 4. Please, at the end of this section, give a clear and direct answer to the question that was launched at the beginning of the paper "Are microalgae new players in nitrous oxide emissions from 2 eutrophic aquatic environments?"

Section 4.4 was added to the manuscript (L292-303):” To this date, it is still unclear whether microalgae contribute significantly to the N2O emissions reported from eutrophic aquatic ecosystems, especially for estuaries and coastal ecosystems. However, microalgae have been showed to synthesise N2O in various settings and preliminary assays suggested their involvement in N2O emissions in lake samples [55]. Further field data and/or microcosms are therefore needed to confirm these findings in many other aquatic ecosystems. Considering the sensitivity of microalgal N2O synthesis, and the seasonal variability in N2O emissions reported in the field [43,74,106], long-term with wide spatial coverage and high sampling frequency monitoring of various microalgae-rich environments are therefore needed to improve the accuracy of N2O emissions from these systems. The use of genomics would also be critical to unravel the occurrence and ecological implications of microalgal N2O synthesis and the potential interplay between microbial N2O biosynthetic pathways in those environments.”